# Levels of persistent organic pollutants (POPs) in the Antarctic atmosphere over time (1980 to 2021) and estimation of their atmospheric half-lives

**Thais Luarte**[1,2,3], **Victoria A. Gómez**[2,3], **Ignacio Poblete-Castro**[4], **Eduardo Castro-Nallar**[3,5,6], **Nicolas Hunneus**[3,7,8], **Marco Molina-Montenegro**[3,9], **Claudia Egas**[6], **Germán Azcune**[10], **Andrés Pérez-Parada**[10], **Rainier Lohmann**[11], **Pernilla Bohlin-Nizzetto**[12], **Jordi Dachs**[13], **Susan Bengtson-Nash**[14], **Gustavo Chiang**[15], **Karla Pozo**[16,17], **and Cristóbal J. Galbán-Malagón**[2,3,18]

[1]Programa de Doctorado en Medicina de la Conservación, Facultad Ciencias de La Vida, Universidad Andrés Bello, Santiago, 8370251, Chile

[2]GEMA, Center for Genomics, Ecology & Environment, Universidad Mayor, Camino La Pirámide, 5750 Huechuraba, Santiago, 8580745, Chile

[3]Anillo en Ciencia y Tecnología Antártica POLARIX, Santiago de Chile, Chile

[4]Biosystems Engineering Laboratory, Department of Chemical and Bioprocess Engineering, Universidad de Santiago de Chile (USACH), 9170022, Santiago, Chile

[5]Departamento de Microbiología, Facultad de Ciencias de la Salud, Universidad de Talca, Campus Talca, Av. Lircay s/n, Talca, 3460000, Chile

[6]Centro de Ecología Integrativa, Universidad de Talca, Campus Talca, Av. Lircay s/n, Talca, 3460000, Chile

[7]Center for Climate and Resilience Research (CR)[2], Santiago, 8370415, Chile

[8]Department of Geophysics, Faculty of Physical and Mathematical Sciences, University of Chile, Santiago, 8370456, Chile

[9]Centro de Estudios Avanzados en Zonas Áridas (CEAZA), Facultad de Ciencias del Mar, Univ. Católica del Norte, Larrondo 1281, Coquimbo, Chile

[10]Departamento de Desarrollo Tecnológico – DDT, Centro Universitario Regional del Este (CURE), Universidad de la República, Ruta 9 y Ruta 15, Rocha, 27000, Uruguay

[11]Graduate School of Oceanography, University of Rhode Island, Narragansett, Rhode Island 02882, USA

[12]NILU – Norwegian Institute for Air Research, P.O. Box 100, Kjeller, 2027, Norway

[13]Department of Environmental Chemistry, IDAEA-CSIC, c/Jordi Girona 18-26, Barcelona, Catalonia, 08034, Spain

[14]Southern Ocean Persistent Organic Pollutants Program, Centre for Planetary Health and Food Security, School of Environment and Science, Griffith University, Nathan, Queensland, 4111, Australia

[15]Center for Sustainable Research & Department of Ecology and Biodiversity, Faculty of Life Sciences, Universidad Andres Bello, Santiago, 8370251, Chile

[16]Facultad de Ingeniería y Tecnología, Universidad San Sebastián, Lientur 1457, Concepción, Chile

[17]RECETOX, Faculty of Science, Masaryk University, Kotlarska 2, Brno, Czech Republic

[18]Institute of Environment, Florida International University, University Park, Miami, Florida 33199, USA

**Correspondence:** Thais Luarte (thaisluarte@gmail.com) and Cristóbal J. Galbán-Malagón (cristobal.galban@umayor.cl)

Received: 19 January 2023 – Discussion started: 2 February 2023
Revised: 24 April 2023 – Accepted: 26 April 2023 – Published:

**Abstract.** Persistent organic pollutants (POPs) are synthetic compounds that were intentionally produced in large quantities and have been distributed in the global environment, originating a threat due to their persistence, bioaccumulative potential, and toxicity. POPs reach the Antarctic continent through long-range atmospheric

transport (LRAT). In these areas, low temperatures play a significant role in the environmental fate of POPs, retaining them for a long time due to cold trapping by diffusion and wet deposition, acting as a net sink for many POPs. However, in the current context of climate change, the remobilization of POPs that were trapped in water, ice, and soil for decades is happening. Therefore, continuous monitoring of POPs in polar air is necessary to assess whether there is a recent re-release of historical pollutants back to the environment. We reviewed the scientific literature on atmospheric levels of several POP families (polychlorinated biphenyls – PCBs, hexachlorobenzene – HCB, hexachlorocyclohexanes – HCHs, and dichlorodiphenyltrichloroethane – DDT) from 1980 to 2021. We estimated the atmospheric half-life using characteristic decreasing times (TD). We observed that HCB levels in the Antarctic atmosphere were higher than the other target organochlorine pesticides (OCPs), but HCB also displayed higher fluctuations and did not show a significant decrease over time. Conversely, the atmospheric levels of HCHs, some DDTs, and PCBs have decreased significantly. The estimated atmospheric half-lives for POPs decreased in the following order: 4,4' DDE (13.5 years) > 4,4' DDD (12.8 years) > 4,4' DDT (7.4 years) > 2,4' DDE (6.4 years) > 2,4' DDT (6.3 years) > $\alpha$-HCH (6 years) > HCB (6 years) > $\gamma$-HCH (4.2 years). For PCB congeners, they decreased in the following order: PCB 153 (7.6 years) > PCB 138 (6.5 years) > PCB 101 (4.7 years) > PCB 180 (4.6 years) > PCB 28 (4 years) > PCB 52 (3.7 years) > PCB 118 (3.6 years). For HCH isomers and PCBs, the Stockholm Convention (SC) ban on POPs did have an impact on decreasing their levels during the last decades. Nevertheless, their ubiquity in the Antarctic atmosphere shows the problematic issues related to highly persistent synthetic chemicals.

## 1   Introduction

Persistent organic pollutants (POPs) are a group of toxic chemicals primarily produced and used by the agricultural, industrial, and household applications during the third industrial revolution (Safe, 1994; Qiu et al., 2004; Jayaraj et al., 2016). In the last 3 decades, studies have reported that POP levels have soared in the environment worldwide, as these chemicals are highly stable and resistant to degradation (Pennington, 2001). This persistence and their hydrophobicity result in POP bioaccumulation within organisms and biomagnification along food webs (Hop et al., 2002; Fisk et al., 2001a, b; Borga and Di Guardo, 2005), where they may elicit toxic effects, such as endocrine disruption, threatening the health of both wildlife and humans (Brown et al., 2014; Bourgeon et al., 2012). Given their detrimental effects, 35 POP substances are currently regulated internationally by the Stockholm Convention (SC), which seeks to reduce and eliminate POP production and use (UNECE, 1999; UNEP, 2001). However, despite regulatory action among SC signatory nations, considerable levels of POPs are still detected in water, atmosphere, biota, and sediments worldwide due to their persistence and potential for long-range transport, as well as their current emission sources (e.g., Vergara et al., 2019; Vasseghian et al., 2021; Avila et al., 2021; Die et al., 2021; Garcia-Cegarra et al., 2021). Of utmost concern is that these toxic pollutants are present in the environmental compartments of regions far from emission sources that have previously been considered pristine areas, including polar regions (Galbán-Malagón et al., 2013a, b, c; Pozo et al., 2017; Wu et al., 2020; Azcune et al., 2022; Xie et al., 2022).

The Antarctic continent is the most remote region from primary sources of POPs (Von Waldow et al., 2010). POPs reach Antarctica mainly through long-range atmospheric transport (LRAT), which generally occurs by the process known as "grasshopping", consisting of successive atmospheric volatilization and depositions (Blais et al., 2007; Brown and Wania, 2008; Bengtson-Nash, 2011; Jurado and Dachs, 2008). Ocean currents also contribute to their transport processes, albeit at longer timescales since the Antarctic Circumpolar Current acts as a barrier limiting the oceanic transport of POPs to the Antarctic continent (Bengtson-Nash, 2011). The "barrier theory" has been questioned by Lozoya et al. (2022) for the South Shetland Islands, where the current experiences topographical forcing through the Drake Passage. Finally, another minor transport process is biological, mediated by migratory biota (Braune et al., 2005; Wild et al., 2022). In addition, there may be local sources of POPs, such as research stations and tourist hotspots, that can contribute to detectable and sometimes elevated concentrations of POPs. For example, PCBs have been reported in the vicinity of such local sources (Larsson et al., 1992; Risebrough et al., 1990; Hale et al., 2008). The low temperatures of Antarctica play an important role in the environmental fate of POPs, repressing re-volatilization processes and favoring cold trapping (Wania and Mackay, 1996; Casal et al., 2019), limiting any potential degradation, and enhancing bioaccumulation. In this context, several studies show that polar regions act as a net sink for many POPs; Antarctica is a vast continent covered in ice surrounded by the Southern Ocean, hence chemicals deposited through LRAT will first deposit in these compartments (Mackay and Wania, 1995; Kallenborn et al., 1998; Dickhut et al., 2005; Gioia et al., 2008; Cincinelli et al., 2009; Baek et al., 2011; Cabrerizo et al.,

2017; Galbán-Malagón et al., 2012, 2013a, c; Montone et al., 2013). For example, there is evidence supporting oceanic sequestration by the biological pump during blooms that bury these compounds on the seafloor (Galbán-Malagón et al., 2013a, c) or by biodegradation due to the microbial loop (Galbán-Malagón et al., 2013d). In the context of rapid climate change experienced in polar regions, the remobilization of POPs previously trapped in water, ice, and soil for decades is expected (Nizzetto et al., 2010; Ma et al., 2011; Cabrerizo et al., 2013). The re-emission of POPs to the environment will affect global efforts to moderate human and environmental exposure to these toxic compounds (Bigot et al., 2016); therefore, continuous monitoring of POP levels in polar abiotic matrices is necessary to assess the extent to which such re-emissions to the atmosphere occur.

The detection of chemicals in remote regions serves as direct empirical evidence of a compound's persistence and potential for long-term environmental transport (Bengston-Nash et al., 2017). POPs were first reported in Antarctic biota in the 1960s (Sladen et al., 1966; Tatton and Ruzicka, 1967), sparking interest in studying the transport, fate, and levels present in different environmental compartments. Through the collation of decades of coordinated monitoring data of POPs in the Arctic atmosphere, studies have explored the fate, sources, and long-range transport of POPs in the Northern Hemisphere (Hung et al., 2010, 2016; Wu et al., 2010, 2011). A general downward trend of many airborne POPs has been demonstrated in the Arctic (Hung et al., 2010, 2016; Kong et al., 2014). However, continuous and consistent atmospheric measurements on POPs in Antarctica are limited, due to the remote geographical location and complex climatic conditions of this continent, which put logistical constraints on any monitoring program. These knowledge gaps make it difficult to understand the fundamental patterns of POPs in this area (Bengston-Nash, 2011) but also facilitate systematic comparison with studies conducted in the Arctic.

This paper presents the first systematic review of the most reported POPs in the Antarctic atmosphere, allowing us to summarize the data collected by the different studies and compare the concentrations recorded over the years and at the different sampling sites. Such a compilation allows the identification of temporal trends and calculate the atmospheric half-lives of the predominant POPs being monitored to provide insights into expected impacts of environmental remobilization under changing Antarctic conditions.

## 2 Methods

### 2.1 Compilation of bibliographical data

We reviewed all published studies on atmospheric levels of the most reported POP families in the Antarctic atmosphere (polychlorinated biphenyls (PCBs), hexachlorobenzene (HCB), hexachlorocyclohexane (HCH), and dichlorodiphenyltrichloroethane (DDT) and its degradation products) from 1980 to 2021. An exhaustive search was performed in the Web of Science and Scopus databases using the words "Persistent Organic Pollutants", "atmospheric", and "Antarctica", including only articles written in English; excluding from the analysis references that do not refer to a good quality assurance and quality control measures. Thus, studies not reporting information about blank samples, limits of detection, limits of quantification, and/or instrumental detection limits, and referring to previous works reporting the quality criteria used, were not included in the present study. This is important because the reported concentrations are at very low levels, and to avoid bias, it is necessary to be sure about the quality measures of sample collection and analysis. A total of 34 publications were found, from which we retrieved data on the levels reported, the year in which the samples were collected, and the sampling sites (Tables 1, 2 and 3). We worked exclusively with the levels of the target compounds in the gas phase, obtained from active and passive sampling. Furthermore, compounds scarcely reported in the Antarctic atmosphere, such as polybrominated diphenyl ethers (PBDEs), polycyclic aromatic hydrocarbons (PAHs), and per- and polyfluoroalkyl substances (PFASs), were excluded.

### 2.2 Statistical analysis

To evaluate the differences between the levels present in West Antarctica and East Antarctica, a non-parametric Mann–Whitney $U$ variance analysis was conducted. To estimate the trend in the change of concentrations, a linear regression was performed between the natural logarithm of the concentrations for each year studied. Finally, a generalized linear model (GLM) was performed to elucidate whether the variability in the atmospheric POP levels reviewed is due to the different types of sampling used by the different studies (active or passive sampling) or the time variable. All the analyses were performed using the R statistical software (R Core Team, 2022).

### 2.3 Estimation of characteristic decreasing times (TD)

Atmospheric half-lives were estimated by deriving the $e$-folding or characteristic decreasing times ($T_D$), following the methodology of Galbán-Malagón et al. (2013a). The half-live is defined as the time needed to decrease the atmospheric concentration by 35 % ($e^{-1}$) of its initial concentration, which is given by 0.69 $T_D$. First, only the studies that reported all the values recorded for each sample were used (Tables S1 and S2). These studies were ordered by year of sampling, and their respective $T_D$ was calculated by least squares adjusting the concentrations to Eq. (1):

$$\ln C_{\text{Atm}} = -k_d t + b, \tag{1}$$

**Table 1.** HCB and HCH levels ($\mathrm{pg\,m^{-3}}$) in the Antarctic atmosphere from 1980 to present. $\sum n$ indicates the number of isomers included in the study. The notation "n.d" indicates "not detected".

| Sampling area | Type of sampling | Year | HCB | $\alpha$-HCH | $\gamma$-HCH | $\sum$HCHs | $\sum n$ | Reference |
|---|---|---|---|---|---|---|---|---|
| Southern Ocean | Active | 1980–1981 | | | | 90–170 | | Tanabe et al. (1982) |
| Southern Ocean | Active | 1981–1982 | | | | 44-170 | | Tanabe et al. (1983) |
| Cape Town and Newmayer Station | Active | 1999 | | 0.36 | 0.15 | | | Lakaschus et al. (2002) |
| Ross Island | Passive | 1988–1999 | | | 25.8 (0.5–118) | | | Larsson et al. (1992) |
| East Antarctica | Passive | 1990 | 62.6 (40–78) | 3.2 (2.8–3.6) | 2.4 (1.1–5.6) | 5.7 | 2 | Bidleman et al. (1993) |
| Signy Island | Active | 1994–1995 | | 2.8 | 21.8 | 26.97 | 3 | Kallenborn et al., 1998 |
| East Antarctica | Passive | 1997–1998 | | 1.06 (0.81–1.4) | | | | Jantunen et al. (2004) |
| Terra Nova Bay | Passive | 1993 | 21 (n.d.–28) | | | 13 (5–20.0) | 3 | Kallenborn et al. (1998) |
| Ross Island | Active | 1995 | (<0.6–25.3) | | | 3.9–32.5 | 2 | Montone et al. (2005) |
| West of the Antarctic Peninsula and southwest of Adelaide Island | Active | 2001–2002 | 19.4 (<5–32.1) | 0.3 (<0.05–0.52) | 0.755 (<0.02–2.98) | | | Dickhut et al. (2005) |
| Terra Nova Bay | Active | 2003–2004 | 11.4 (6.0–20) | | | 0.8 (0.3–1.2) | 2 | Gambaro et al. (2005) |
| Terra Nova Bay | Active | 2003–2004 | 11.4 (5.93–20.4) | | | 0.22 (0.1–0.35) | 2 | Cincinelli et al. (2009) |
| Ny-Ålesund, King George Island, and Chuuk | Passive | 2005–2009 | | | | | | Baek et al. (2011) |
| South Scotia | Active | 2008 | 8.1 (2.18–15.82) | 1.7(0.06–5.84) | 4.6 (1.5–7.1) | | | Galbán-Malagón et al. (2013b) |
| Weddell | Active | 2009 | 19.5 (2.4–30.1) | 0.16 (0.05–2.09) | 0.84 (0.1–1.87) | | | Galbán-Malagón et al. (2013b) |
| Bransfield Sea | Active | 2009 | 16.7 (3.3–34.24) | 0.14 (0.04–0.46) | 1.15 (0.2–3) | | | Galbán-Malagón et al. (2013b) |
| Bellingshausen | Active | 2009 | 42.9 (27.31–49.71) | 0.26 (0.22–0.16) | 0.14 (0.07–0.19) | | | Galbán-Malagón et al. (2013b) |
| Palmer Station | Active | 2010 | 34 (26.2–37.7) | 0.81–1.68 | 0.87–2.31 | | | Khairy et al. (2016) |
| Troll Station/ Queen Maud Land | Active | 2010 | 22.9 | | | | | Kallenborn et al. (2013) |
| Ross Sea | Passive | 2010–2011 | 22.8 (0.8–50) | 0.5 (n.d.–0.5) | n.d. | 0.5 (n.d.–0.5) | 2 | Pozo et al. (2017) |
| Antarctic Plateau | Active | 2011 | (0.67–2.7) | | BD-2.7 | | | Cabrerizo et al. (2017) |
| Antarctic marginal seas | Active | 2013–2014 | 2.6 (0.081–10) | | | (n.d.–6.8) | 3 | Wu et al. (2020) |
| Southern Ocean between Australia and Antarctica | Active | 2014 | (<22–35) | <0.13–1.1 | <0.70–4.3 | n.d.–3.65 | 3 | Bigot et al. (2016) |
| King George Island | Passive | 2012–2018 | 163 (99.2–252) | 1.4 (0.5–13.6) | 0.1–7.9 | 0.7–22.3 | 4 | Hao et al. (2019) |

where $k_d$ is the inverse of the $e$-folding time $T_D$ (in years), $t$ is the time in years, and $b$ is the independent term; $T_D$ was not calculated for $\beta$-HCH, due to the limited data available.

## 3 Results and discussion

### 3.1 Organochlorine pesticides (OCPs)

Organochlorine pesticides (OCPs) represent most of the POPs listed in the Stockholm Convention. These organic compounds have been widely produced and commercialized since the 1950s for agricultural use and vector control (UN-ECE, 1999; UNEP, 2001). The application of technical HCH in agriculture has been banned since the early 1980s, while DDT, Lindane ($\gamma$-HCH), and HCB were banned in the 1990s (UNECE, 1999; UNEP, 2001). The OCPs were first reported in Antarctic marine biota in the late 1960s by Sladen et al. (1966) and Tatton and Ruzicka (1967). To date, their levels in different environmental compartments continue to be

**Table 2.** DDT levels (pg m$^{-3}$) in Antarctic atmosphere from 1988 to present.

| Sampling area | Type of sampling | Year | 2,4 DDE | 4,4-DDE | 2,4DDD | 4,4 DDD | 2,4 DDT | 4,4 DDT | Reference |
|---|---|---|---|---|---|---|---|---|---|
| Ross Island | Passive | 1988–1999 | | 1 | | | | 2 | Larsson et al. (1992) |
| East Antarctica | Passive | 1990 | | | | | | 0.53 | Bidleman et al. (1993) |
| Signy Island | Active | 1994–1995 | 0.07 | 0.4 | 0.068 | 0.098 | 0.195 | 0.2 | Kallenborn et al. (1998) |
| Ross Island | Active | 1995 | | 9.2 | | 11.7 | | 8.1 | Montone et al. (2005) |
| Antarctic marginal seas | Active | 2013–2014 | 0.097 | 0.35 | 0.043 | 0.034 | 0.17 | 0.12 | Wu et al. (2020) |
| Southern Ocean between Australia and Antarctica | Active | 2014 | <0.51 | <0.15–0.44 | <1.6 | <1.8 | <2.7 | <7.8 | Bigot et al. (2016) |
| King George Island | Passive | 2012–2018 | 0.2 | 0.6 | 0.1 | 0.2 | 0.1 | 0.24 | Hao et al. (2019) |

reported (e.g., Vergara et al., 2019; Wu et al., 2020; Krasnobaev et al., 2020; Xie et al., 2022).

### 3.1.1 Atmospheric levels of organochlorine pesticides (OCPs)

In the Arctic atmosphere, HCB concentrations are the highest of any OCPs (De March et al., 1998). Similarly, atmospheric concentrations of HCB reported from the Antarctic have been observed to be higher than the other target OCPs (Tables 1 and 2), being the most frequently detected and abundant POP in the Antarctic atmosphere (Kallenborn et al., 2013; Wang et al., 2018; Hao et al., 2019; Wu et al., 2020). Temporal patterns of atmospheric HCB concentrations in the Antarctic show significant interannual fluctuations with low but significant decreasing trend ($p < 0.001$, See Table 1), with a higher variability over time,specially in the last decade (Fig. 1a). A clear decrease in concentrations is shown until about 2010; thereafter, a large variability of data is shown where the trend seems to be changing. However, there is a lack of sufficient data to be able to confirm this trend. The maximum values were reported by Hao et al. (2019) during the 2012–2018 sampling period on King George Island (Table 1). Such increases in HCB gaseous levels could be mainly associated with re-emission from environmental surfaces (water, soil, and snow) shifting from a reservoir to a secondary source of this compound on the Antarctic continent. The HCB is the most persistent OCP chemical assessed here, as suggested before (Galbán-Malagón et al., 2013b). In addition, there may still be an important influence of transport from current primary sources (i.e., combustion and thermal processes) on a global scale and unintentional formation during thermal processing or combustion of chlorine-containing materials (Barber et al., 2005). The trend shown in Fig. 1a points out that concentrations of HCB in the Antarctic atmosphere may be regionally dependent and may be high due to the climate and/or environmental change processes occurring in different Antarctic regions.

The reported atmospheric concentrations of ΣHCHs in Antarctica from 1980 to 2019 show a decreasing trend over time (Table 1; Fig. 1b and c), with significant differences in interannual levels ($P < 0.05$). The maximum concentration of HCHs was 170 pg m$^{-3}$, reported in 1980–1982 (Tanabe et al., 1982, 1983), and progressively lower concentrations reaching values under detection levels and below 1 pg m$^{-3}$ are reported from 2003 to 2019 (Gambaro et al., 2005; Cincinelli et al., 2009; Baek et al., 2011; Galbán-Malagón et al., 2013b; Kallenborn et al., 2013; Pozo et al., 2017; Cabrerizo et al., 2017; Wu et al., 2020; Bigot et al., 2016; Hao et al., 2019). The $\gamma$-HCH isomer was found at high concentrations in Antarctica between 1989 and 1990, with a maximum atmospheric concentration of 118 pg m$^{-3}$ in 1988 at Ross Island, by Larsson et al. (1992) (Fig. 1c, Table S1). Decreasing concentrations are then reported for $\gamma$-HCH in 2000, which is unsurprising if fresh sources have been removed, given the lower volatility and higher water solubility of this isomer. On the other hand, the $\alpha$-HCH isomer, has been increasing since 2006 (Baek et al., 2011; Galbán-Malagón et al., 2013b; Hao et al., 2019), compared to the concentrations recorded during 2001–2004 by Dickhut et al. (2005) and Cincinelli et al. (2009).

Published studies reporting gaseous levels for DDT and their isomers from 1988–2021 were lower than the rest of the target OCPs, and like HCHs, the DDTs showed a decreasing trend over the years (Table 2, Fig. 2), with significant interannual differences ($p < 0.05$) for compounds 4,4'-DDT, 4,4'-DDE, 2,4'-DDT, and 2,2'-DDE, and non-significant annual differences ($p > 0.05$) for compounds 4,4'-DDD and 2,4'-DDD.

To date, atmospheric concentrations of HCB, $\alpha$-HCH, $\beta$-HCH, $\gamma$-HCH, 2,4'-DDTs, 4,4'-DDTs, and 2,4' DDD isomers have been studied over much of the Antarctic continent, both in West Antarctica (Kallenborn et al., 1998; Montone et al., 2005; Dickhut et al., 2005; Baek et al., 2011; Galbán-Malagón et al., 2013c; Khairy et al., 2016; Hao et al., 2019) and in East Antarctica (Tanabe et al., 1982, 1983; Lakaschus et al., 2002; Larsson et al., 1992; Jantunen et al., 2004; Bidleman et al., 1993; Gambaro et al., 2005; Cincinelli et al., 2009; Kallenborn et al., 2013; Pozo et al., 2017; Cabrerizo et al.,

**Table 3.** PCB levels (pg m$^{-3}$) in the Antarctic atmosphere from 1988 to present. $\sum n$ indicates the number of congeners included in the study. The notation "n.d." indicates "not detected".

| Sampling area | Type of sampling | Year | $\sum$PCBs | $\sum$n | Reference |
|---|---|---|---|---|---|
| Ross Island | Passive | 1988–1990 | 15.2 | 6 | Larsson et al. (1992) |
| King George Island | Active | 1993–1994 | 20.8 (12.09–42.8) | 10 | Montone et al. (2003) |
| Signy Island | Active | 1994–1995 | (0.01–17.2) | 22 | Kallenborn et al. (1998) |
| Ross Island | Active | 1995 | 62.4 | 11 | Montone et al. (2005) |
| King George Island | Active | 1996–1996 | 37.4 (12.1–92.6) | 10 | Montone et al. (2003) |
| Terra Nova Bay | Active | 2003–2004 | 1.06 (0.61–1.78) | 61 | Gambaro et al. (2005) |
| Ny-Ålesund, King George Island, and Chuuk | Passive | 2005–2009 | 60.3(22.8–87.1) | 11 | Baek et al. (2011) |
| Ny-Ålesund, King George Island, and Chuuk | Passive | 2005–2009 | 19.8 (11.1–31.9) | 205 | Baek et al. (2011) |
| ICEPOS | Active | 2005 | 16.84 (7.12–25.65) | 25 | Galbán-Malagón et al. (2013c) |
| South Scotia sea | Active | 2008 | 45.13 (6.2–78.9) | 25 | Galbán-Malagón et al. (2013c) |
| Antarctic Peninsula | Active | 2009 | 12.13 (1.8–38.1) | 25 | Galbán-Malagón et al. (2013c) |
| Polish beach | Active | 2009 | (2.1–3.1) | 25 | Galbán-Malagón et al. (2013c) |
| Livingston Island | Active | 2009 | 7.23 (3.5–12.9) | 25 | Galbán-Malagón et al. (2013c) |
| King George Island | Passive | 2009–2010 | 1.142 | 7 | Li et al. (2012b) |
| King George Island | Passive | 2009–2010 | 36.837 | 19 | Li et al. (2012a) |
| King George Island, Antarctica. | Passive | 2009–2010 | 4.34 | 7 | Li et al. (2012b) |
| Troll Station/Queen Maud Land | Active | 2010 | 0.5 | 32 | Kallenborn et al. (2013) |
| Palmer Station | Active | 2010 | 12 | 29 | Khairy et al. (2016) |
| Ross Sea | Passive | 2010–2011 | 0.46 (0.14–1.13) | 7 | Pozo et al. (2017) |
| Antarctic Plateau | Active | 2011 | (0.8–27) | 26 | Cabrerizo et al. (2017) |
| King George Island | Active`TS1` | 2011–2014 | 5.39 (0.91–35.9) | 7 | Wang et al. (2015) |
| King George Island | Active | 2011–2014 | 5.87–72.7 (26.1)`TS2` | 20 | Wang et al. (2017) |
| King George Island | Passive | 2010–2018 | 10.4 (1.5–29.7) | 19 | Hao et al. (2019) |
| Antarctic marginal seas | Active | 2013–2014 | 1.1 (n.d.–6.7) | 14 | Wu et al. (2020) |

2017; Wu et al., 2020; Bigot et al., 2016). The detected concentrations of HCB, $\alpha$-HCH, and 4,4'-DDT indicate significant spatial differences ($P < 0.05$), with higher atmospheric concentrations in West Antarctica than in East Antarctica (Table S4). The $\gamma$-HCH and 2,4'-DDT isomers did not show spatial differences between the two zones ($P > 0.05$) (Table S4), but the usage of these compounds decreased in a similar way from 1990 to 2000 (Vijgen, 2006). This can be explained by two causes together. The first is the greater proximity of South America to the Antarctic Peninsula. The proximity itself has to do with the possibility of transport of these compounds from southern South America where it is suggested that air samples influenced by the continent are capable of transporting pollutants from South America to Antarctica (Dickhut et al., 2005) such as Heptachlor epoxide. However, when looking to usage reported in South America compared to Africa (Li, 1999), this could influence the abundance of $\alpha$-HCH in the western Antarctic area. Examining previous information for both HCB and 4,4'-DDT, there is not a great deal of information about the use of these compounds in areas near Antarctica, but the proximity to South America could explain these variations in conjunction with

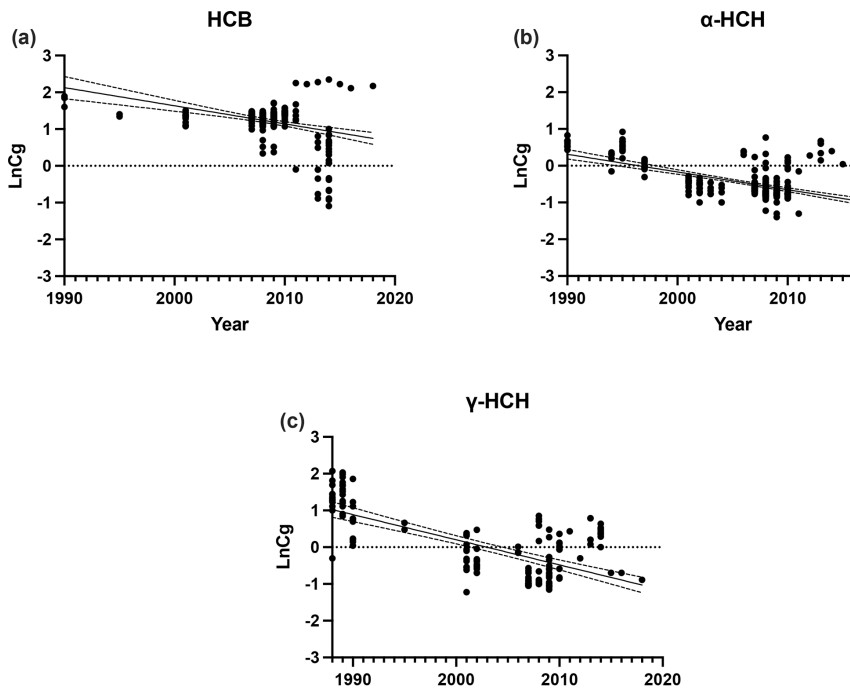

**Figure 1.** Atmospheric levels ($\mathrm{pg\,m^{-3}}$) of **(a)** HCB, **(b)** $\alpha$-HCH, and **(c)** $\gamma$-HCH over time.

the paucity of data in East Antarctica. On the other hand, the Mann–Whitney $U$ variance analysis was not performed for the $\beta$-HCH and 2,4'-DDD isomers, because all levels reported in East Antarctica were below the detection limit. In addition, the results of the generalized linear model indicate that the variability of atmospheric OCPs is mainly due to the year variable, with no significant differences ($p>0.05$) between the atmospheric levels obtained from active and passive sampling (see Table S5).

## 3.1.2 Atmospheric half-lives of organochlorine pesticides (OCPs)

The half-life was estimated for all OCP compounds with significant interannual differences ($p<0.05$ or lower). The estimated half-lives decreased in the following order: 4,4'-DDT (17.2 years) > 2,4'-DDT (14.4 years) > $\alpha$-HCH (14.3 years) > HCB (14.0 years) > $\gamma$-HCH (10.1 years) (more details are given in Table 4). The higher atmospheric half-life values estimated in this study for DDTs isomers, compared to the values estimated for HCHs and HCB, might be related to the years in which these compounds were banned, since DDTs were banned approximately 10 years after HCHs isomers. It may also be due to continuous production and use of DDTs in some parts of the world due to exemptions to the Stockholm Convention. The estimated values are higher than the atmospheric half-lives reported by other authors, such as Atkinson (1986); Howard (1991); Mortimer and Connel (1995); and Kelly et al. (1994), whose

estimated and published values do not exceed 1 year. However, the methodologies employed differ from the one used in the present study where Atkinson (1986); Howard (1991); and Mortimer and Connel (1995) were based on rate constant of gas-phase reaction with OH radical for trichlorobiphenyls, while Kelly et al. (1994) were based on atmospheric transformation lifetime. On the other hand, if we compare studies with similar methodology, the study by Venier and Hites (2010) in Great Lakes shows that the half-life estimates for $\alpha$-HCH and $\gamma$-HCH are in a similar range to our estimates, while the one obtained for 4,4'-DDT is slightly lower (Fig. 4a). Similarly, according to the half-life estimates by Wong et al. (2021), HCB shows higher values than those reported by us, but they report similar values than ours for $\alpha$-HCH, $\gamma$-HCH, 2,4'-DDT, and 4,4'-DDT (Fig. 4a).

Polar areas are often considered to be a net sink for POPs. Studies have documented that $\alpha$-HCH and $\gamma$-HCH exchanges preferentially from air to water, with this diffusion being the predominant atmospheric deposition mechanism (Galbán-Malagón et al., 2013a, c; Dickhut et al., 2005; Cincinelli et al., 2009; Jantunen et al., 2004; Lohmann et al., 2009; Xie et al., 2011; Zhang et al., 2012; Huang et al., 2013). Once deposited onto surface waters, they are susceptible to sequestration by the biological pump (Galbán-Malagón et al., 2013a, c), as well as to degradation driven by hydrolysis and biodegradation to a minor extent (Harner et al., 2000; Helm et al., 2002; Galbán-Malagón et al., 2013c). These processes minimize the opportunity for re-entry into the atmosphere through volatilization. The lower half-life

**Table 4.** Estimated atmospheric half-lives of examined POP compounds.

| Compounds | $T_{1/2}$ (years) | 95 % confidence interval | $R^2$ | $p$ value | Equation |
|---|---|---|---|---|---|
| HCB | 14.0 | 10.6–20.7 | 12.03 | <0.0001 | LnCg $= -0.04931 \times$ year $+ 100.2$ |
| $\alpha$-HCH | 14.3 | 12.4–17.0 | 32.55 | <0.0001 | LnCg $= -0.04817 \times$ year $+ 96.17$ |
| $\gamma$-HCH | 10.1 | 8.6–12.3 | 44.63 | <0.0001 | LnCg $= -0.06837 \times$ year $+ 136.9$ |
| 4,4' DDT | 17.2 | 11.8-31.7 | 23.54 | <0.0001 | LnCg $= -0.04015 \times$ year $+ 79.50$ |
| 2,4 DDT | 14.4 | 9.8–27.3 | 37.55 | <0.001 | LnCg $= -0.04794 \times$ year $+ 94.99$ |
| 2,4 DDE | 17.6 | 9.2–232 | 15.44 | <0.05 | LnCg $= -0.03916 \times$ year $+ 77.93$ |
| PCB 28 | 3.9 | 3.2–5.2 | 43.08 | <0.0001 | LnCg $= -0.1748 \times$ year $+ 351.2$ |
| PCB 52 | 3.7 | 3.2–4.3 | 63.53 | <0.0001 | LnCg $= -0.1887 \times$ year $+ 378.7$ |
| PCB 101 | 4.7 | 4.0–5.6 | 67.42 | <0.0001 | LnCg $= -0.1480 \times$ year $+ 295.8$ |
| PCB 118 | 3.6 | 3.0–4.3 | 55.91 | <0.0001 | LnCg $= -0.1930 \times$ year $+ 385.8$ |
| PCB 138 | 6.5 | 5.3–8.3 | 40.7 | <0.0001 | LnCg $= -0.1066 \times$ year $+ 212.7$ |
| PCB 153 | 7.6 | 6.0–10.4 | 31.59 | <0.0001 | LnCg $= -0.09071 \times$ year $+ 181.2$ |
| PCB 180 | 4.6 | 3.3–8.0 | 24.64 | <0.0001 | LnCg $= -0.1486 \times$ year $+ 296.2$ |

values for HCHs may be related to their lower Henry's law constant (HLC) when compared to other POPs. On the contrary, to our knowledge, no degradation processes have been documented for HCB in surface water, and, furthermore, conditions close to air–water equilibrium have been reported for this compound (Cincinelli et al., 2009; Galbán-Malagón et al., 2013c). Similarly, DDTs are more hydrophobic with much higher $K_{OW}$ values than HCHs (Table S3), so they are rapidly removed from seawater as particles sink (Lohmann et al., 2007). Thus, it is possible that the high half-lives estimated for DDTs and their metabolites DDD and DDE may be due to unknown current primary and secondary sources (Voldner and Li, 1995; Channa et al., 2012, Li et al., 2020).

## 3.2 Polychlorinated biphenyls (PCBs)

Like OCPs, polychlorinated biphenyls (PCBs) were among the first groups of POPs to be listed under the Stockholm Convention and are characterized by their high chemical stability. Prior to their regulatory control in the 1970s, commercial mixtures of PCBs were widely used in many industrial applications, such as fluids in transformers and capacitors, hydraulic fluids, lubricating oils, and as additives in pesticides, inks and paints, flame retardants, plasticizers, sealants for wood and cement surfaces, among others (Kennish, 2017; FAO/UNEP 1992).

The PCBs were first reported in Antarctica in the 1960s and 1970s (Risebrough et al., 1968, 1976), and since then, numerous studies have reported their levels in air, water, sediments, snow, and biota on the Antarctic continent (e.g., Kallenborn et al., 1998; Fuoco et al., 1995; Gupta et al., 1996; Weber et al., 2003; Kim et al., 2015). Here, we selected seven indicator PCB congeners (28, 52, 101, 118, 138, 153, and 180) considering that they are the most reported PCB congeners worldwide, including Antarctica.

### 3.2.1 Atmospheric levels of polychlorinated biphenyls (PCBs)

The atmospheric concentrations of $\Sigma_7$PCBs reported by the reviewed studies were below those of the target OCPs (Table 3). Overall, the levels of $\Sigma_7$PCBs reported from 1980 to 2021 showed a decreasing trend over time (Tables 3 and 4, Fig. 3), with significant differences in their levels ($p < 0.05$). Congeners 28 and 52 recorded the highest concentrations on King George Island, with values of 69.9 pg m$^{-3}$ in 1995, and 33.2 pg m$^{-3}$ in 1996, reported by Montone et al. (2005, 2003) (Fig. 3a and b, Table S2). In contrast, the lowest concentrations of all target PCBs were reported for congener 180, ranging from not detected (n.d.) to 3.4 pg m$^{-3}$ (Fig. 3g, Table S2).

Like OCPs, atmospheric concentrations of the seven PCB congeners have been reported over most of the Antarctic zone, covering the zone in West Antarctica (Montone et al., 2003, 2005; Kallenborn et al., 1998; Baek et al., 2011; Galbán-Malagón et al., 2013c; Li et al., 2012a, b; Khairy et al., 2016; Wang et al., 2017; Hao et al., 2019; Wu et al., 2020) and East Antarctica (Larsson et al., 1992; Gambaro et al., 2005; Kallenborn et al., 2013; Pozo et al., 2017; Cabrerizo et al., 2017). Significant spatial differences ($p < 0.05$) were observed in the atmospheric concentrations of congeners 28, 52, 101, and 138, with higher concentrations in West Antarctica than East Antarctica, while there was no significant difference among sites for congeners 101, 118, and 153 ($p > 0.05$). These differences are consistent with the different atmospheric patterns over the Antarctic peninsula regions, with entrance of air masses from the north, and more permanent wet deposition events by snow and rain, increasing the regional concentrations of POPs (Casal et al., 2019; Casas et al., 2021). On the other hand, it is essential to highlight that the variability of PCBs reported in this study is substantially due to the time variable ($p < 0.05$), with no significant differences ($p > 0.05$) between the atmospheric levels of PCBs obtained from active and passive sampling (see Table S5).

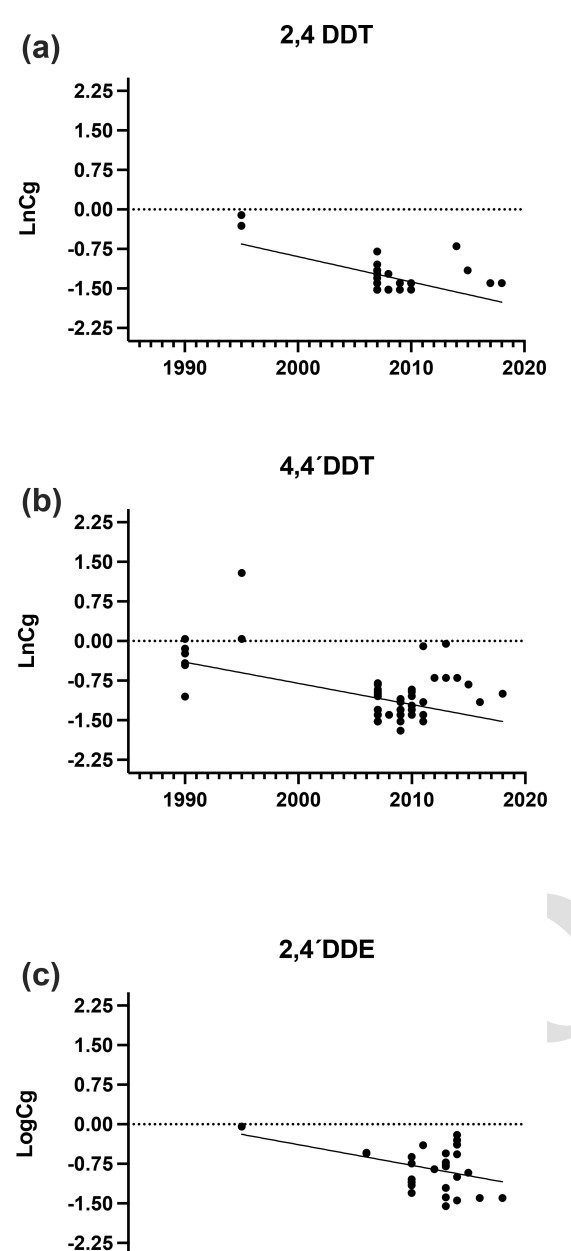

**Figure 2.** Atmospheric levels ($\mathrm{pg\,m^{-3}}$) of **(a)** 2,4'-DDT, **(b)** 4,4'-DDT, and **(c)** 2,4'-DDE over time.

### 3.2.2 Atmospheric half-lives of polychlorinated biphenyls (PCBs)

The estimated atmospheric half-lives for target PCBs decreased in the following order: PCB 153 (7.6 years) > PCB 138 (6.5 years) > PCB 101 (4.7 years) > PCB 180 (4.6 years) > PCB 28 (3.9 years) > PCB 52 (3.7 years) > PCB 118 (3.6 years) (Table 4). The estimated half-lives were directly proportional to the congener's

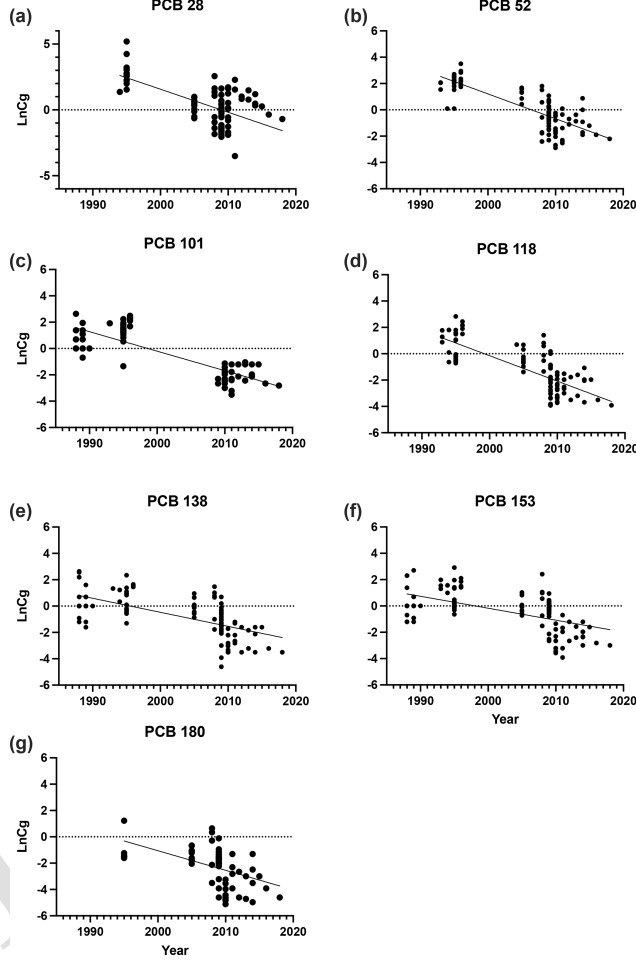

**Figure 3.** Atmospheric levels ($\mathrm{pg\,m^{-3}}$) of **(a)** PCB-28, **(b)** PCB-52, **(c)** PCB-101, **(d)** PCB-118, **(e)** PCB-138, **(f)** PCB-153, and **(g)** PCB-180 over time.

Henry's law constant (HLC) values. (Table S3). Studies by Atkinson (1986) and Sinkkonen and Paasivirta (2000) reported half-lives lower than those estimated in the present work, where none of the estimated half-lives for these compounds exceeded 1 year. However, the methodology of both studies differs from that of the present study, calculating the half-lives of the compounds by means of the rate constant of gas-phase reaction with the OH. Regarding studies using a similar methodology, the atmospheric half-lives estimated by Venier and Hites (2010) in the Great Lakes (United States and Canada) and by Wong et al. (2021) in the Arctic were higher relative to our results for PCBs 28, 52, 101, and 118. They were in a similar range for PCBs 138, 153, and 180 (Fig. 4b).

Studies have documented that the biological pump is highly efficient for PCBs with high hydrophobicity, i.e., high $K_{\mathrm{OW}}$ values (Table S3) (Dachs et al., 2002; Galbán-Malagón et al., 2012, 2013a), thus reducing their re-volatilization. The estimated atmospheric half-lives, however, do not re-

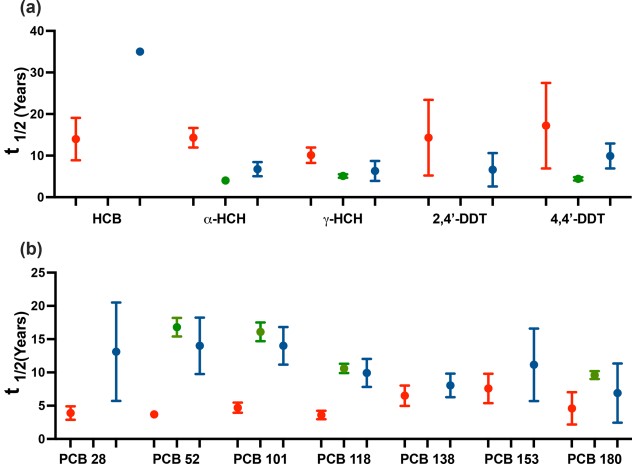

**Figure 4.** Comparison among the estimated atmospheric half-lives obtained in the present work (green) compared with similar estimations from the Great Lakes (red; Venier and Hites, 2010) and the Arctic (blue; Wong et al., 2021) for **(a)** organochlorine pesticides (HCB, $\alpha$-HCH, $\gamma$-HCH, 2,4'-DDT, and 4,4'-DDT) and **(b)** polychlorinated biphenyls (28, 52, 101, 118, 138, 153, and 180).

flect lower values for the compounds with higher $K_{OW}$ (e.g., PCB 138, 153 and 180), so other factors may be influencing the high estimated half-lives of the more hydrophobic PCBs. One of these factors could be the presence of local sources of certain PCB congeners, since it has been reported that higher PCB concentrations are monitored near the research stations compared to sites farther away from these stations, specifically PCB congeners 28, 52, 56, and 101 (Li et al., 2012b; Montone et al., 2003). Furthermore, remobilization of PCBs stored in soils and ice (Cabrerizo et al., 2013; Casal et al., 2019) could be another factor modulating the surface, and thus atmospheric, concentrations of POPs.

### 3.3 Influence of global climate change on the dynamics of POPs in the Antarctic continent

Over the past decades, global climate changes and the effects of increasing temperatures have been observed in the Northern and Southern hemispheres (Hung et al., 2022). Increases in ambient temperature can influence physical and chemical processes and ecosystem changes. For example, it has been reported that increasing ambient temperature will affect the dynamics and exchange of POPs between different environmental matrices. Some studies have exposed the relationship between climate change and POP concentrations (Vorkamp et al., 2022; Potapowicz et al., 2018), describing how POPs are temporarily stored in sediments/soils and can be released into the environment with thawing permafrost (Potapowicz et al., 2018) and that there is an increase in POP availability following iceberg calving (Vorkamp et al., 2022) or increased soil remobilization by up to 45 % (Cabrerizo et al., 2013). In addition, several studies show that seawater, snow, and presumably Antarctic soil are becoming critical secondary sources for POP remobilization (Cabrerizo et al., 2012, 2013; Klanová et al., 2008; Casal et al., 2019).

On the other hand, the mean annual air temperature along the western Antarctic Peninsula has been reported to have increased by as much as 3.4 °C. In addition, the mid-winter temperature increased by 6.0 °C over the past 50 years, making the region one of the most critically affected by climate change (Vaughan et al., 2003; Turner et al., 2005). However, evidence from field sampling indicated that, to date, there is no relationship between atmospheric ambient temperature and atmospheric concentrations of HCH and HCB in East Antarctica (Bengtson-Nash et al., 2017). On the other hand, increasing ambient temperature leads to decreased snow cover, nutrient runoff from land to sea, and increased bioavailability of nutrients on land, causing an increase in primary producers on land and sea (Wasley et al., 2006). In this context, it has been demonstrated in aquatic systems that an increase in primary productivity is vital in the sedimentation processes of POPs from the surface to the aquatic bottom through the biological pump (Larsson et al., 2000; Galbán-Malagón et al., 2018). This process contributes mainly to the removal of POPs from the environment, despite the adverse effects of an increase in primary productivity in ecosystems (e.g., increase in organic matter). Thus, global warming in Antarctica not only implies a temperature change but also leads to multiple processes that can affect the biogeochemical dynamics of POPs, which plays an essential role in the environmental fate of POPs. Depending on the future effects of climate change, Antarctica can act as a secondary source of POPs through the re-volatilization of these compounds or as a sink, contributing to the decrease of their environmental levels. Therefore, future exploration of the impact of climate change is necessary and establishes the importance of establishing long-term monitoring networks.

### 3.4 Potential sources of bias

As presented here, several factors can be considered as sources of bias from historical data analysis. First, in the time frame of this study (1980–2021), analytical instrumentation and laboratory techniques exhibited dramatic change, particularly with the advent of advanced mass spectrometry (MS) over electron capture detection (ECD) or novel calibration techniques based on isotopically labeled standards (Azcune et al., 2022). Therefore, recent data are generated by more sophisticated techniques and modern laboratory QA/QC criteria. On the other hand, we also included studies using active and passive sampling, but no major differences in the values obtained were observed (See Tables 1, 2, and 3) for the whole compounds which agrees with intercalibration experiments conducted in other areas comparing passive and active sampling together (Prats et al., 2022). The published information from Antarctica is reduced to a group of individual experiences in different geographical locations of in-

ternational teams working in the field under different conditions and levels of competence that are difficult to obtain and analyze. One might expect that studies that show a strong track record in Antarctic research, reporting POP levels over a time series, might have greater validity due to constancy and consistency in both sampling and the types of analyses used (e.g., Larsson et al., 1992; Baek et al., 2011; Hao et al., 2019).

In summary, the source of bias, related to the technological advancement of the analyses of the collected samples, could have relevance in the observed variability of the historical trends of HCB and HCH (see Fig. 1a to c). In these cases, it is suggested to continue with dedicated monitoring of these POPs in the coming years to obtain robust observations and conclusions on the degradation of POPs in the Antarctic atmosphere.

## 4 Conclusion

In the present review, a clear trend of decreasing concentrations of PCBs and most targeted OCPs in the Antarctic atmosphere from 1980 to 2021 is documented. This is in response to the hypothesis raised historically about the decrease in atmospheric levels of historical POPs (Vecchiato et al., 2015). However, it opens the door to study new families of pollutants for which there is already analytical capacity that was not available in previous decades. In the case of HCH isomers, DDT, and PCB congeners, high atmospheric concentrations were reported for the 1990–1999 decade, but these compounds were highly restricted since the 1970s. After that date, a strong decrease was observed in the Antarctic atmosphere, which shows that the Stockholm Convention ban on POPs did have the intended impact on the (atmospheric) concentrations over time. However, these compounds are still ubiquitous in the Antarctic atmosphere with atmospheric half-lives of more than 3 years. On the other hand, the revised atmospheric levels of HCB show a decrease in the decade of its prohibition (1990). However, from the year 2000 onwards, they show strong fluctuations in the literature, with values even higher than those reported in 1990. It is noteworthy to consider that a decrease of the atmospheric concentrations does not imply a decrease of the total POPs in Antarctica, an issue that will require future work. In fact, the re-emission of HCB and other POPs from environmental surfaces, such as water, soil, and snow, is a product of its high stability in the environment that has not been deeply studied and could represent a potential source of bias for future works. Studies to date in Antarctica do not allow conclusions to be drawn about the influence of temperature on the environmental fate of POPs on the Antarctic continent. This is due to the lack of consistent time series data as historically conducted in the Arctic. Moreover, our results point to the importance of periodic monitoring and the need to establish monitoring networks with continuous sampling campaigns, not only with aim of monitoring the legacy of POPs but also to identify new pollutants that have the potential to reach Antarctica (e.g., new flame retardants, per- and polyfluoroalkyl substances (PFAS), and polycyclic aromatic hydrocarbons (PAHs), among others). There is increasing evidence of the presence of emerging compounds in different environmental matrices in Antarctica. However, the current surveillance of atmospheric pollutants is related to specific research groups, instead of coordinated efforts between countries with Antarctic presence, where continuous monitoring networks could be generated with the inclusion of various persistent toxic chemicals, as analogous to the efforts done by the Arctic Monitoring and Assessment Program (AMAP) or the Integrated Atmospheric Deposition Network (IADN) in the Great Lakes. In this sense, to establish a monitoring program for assessment of POP levels in the Antarctic atmosphere will depend on the capabilities and the facilities, since an active sampling strategy will benefit from a higher resolution in the assessment of POP trends in the monitoring points, but on the other hand, the use of passive sampling strategy could represent a high spatial coverage to monitor trends but lower time resolution. However, there is need to establish a bigger coordinated monitoring network in the future following the previous experience gained from AMAP.

**Data availability.** All the data used in the present study are reported in the Supplement.

**Supplement.** The supplement related to this article is available online at: https://doi.org/10.5194/acp-23-1-2023-supplement.

**Author contributions.** TL and CJGM designed the research, retrieved the data, performed the analysis, and drafted the original version of the paper. VAG, IPC, ECN, NH, MMM, CE, GA, APP, RL, PBN, JD, SBN, GC, and KP participated in the writing of the submitted version of the paper, commenting during the whole process. All the authors participated in the writing of the final version of the paper.

**Competing interests.** The contact author has declared that none of the authors has any competing interests.

**Disclaimer.** Publisher's note: Copernicus Publications remains neutral with regard to jurisdictional claims in published maps and institutional affiliations.

**Acknowledgements.** This study was funded by ANID/FONDE-CYT/Iniciación 11150548, ANID/FONDECYT/Regular 1161504, ANID/FONDECYT/Regular 1210946, ANID/PCI REDI170292, ANID-PIA-INACH-ACT192057, INACH REGULAR RT_12_17

(Cristóbal Galbán-Malagón). Thais Luarte acknowledges the funding of the Universidad Andrés Bello office of Doctorate Program Support through the PhD Grant Program and INACH DG_02_21.

**Financial support.** Cristobal J. Galbán-Malagón received funding from ANID through the ANID grants FONDECYT 11150548, 1210946, ANID-PCI-REDI170292, ANID-PIA Anillo-INACH-ACT192057, and INACH grant INACH-REGULAR RT_12_17. Thais Luarte's PhD grant was provided by the "Vicerrectoria de Investigación, Universidad Andrés Bello" and INACH grant DG_02_21. Claudia Egas is funded by the INACh grant DG_03_21. Victoria A. Gómez is funded by ANID through the grant ANID FONDECYT POSTDOC 3230076.

**Review statement.** This paper was edited by Ralf Ebinghaus and reviewed by two anonymous referees.

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

## Remarks from the typesetter