# Peer review of "Levels of persistent organic pollutants (POPs) in the Antarctic atmosphere over time (1980 to 2021) and estimation of their atmospheric half-lives."

_Atmospheric Chemistry and Physics, 2023_

## Referee Comment (RC1)

Atmospheric Chemistry and Physics, ms -acp-2023-25

General Comments:

The manuscript "Levels of persistent organic pollutants (POPs) in the Antarctic atmosphere over time (1980) and estimation of their atmospheric half-lives" by Luarte et al. is well written and of high quality. It reviews scientific literature from 1988 to 2021 that published atmospheric levels of different POPs families. POP families that were studied are polychlorinated biphenyls, hexachlorobenzene, hexachlorocyclohexanes and DDT. With concentration levels published, the respective half-lives were estimated by using characteristic decreasing times. Concentrations levels and half-lives are than compared between the POP families and discussed with their dates of ban under the Stockholm Convention.

The detection of POPs in the remote and vulnerable polar environments has been increasingly discussed during the last years. While a lot of studies on the Arctic environment is available, data on POPs occurrence in the Antarctic environment are rare and highly needed. Therefore, this systematic review of POPs in the Antarctic atmosphere provides a good data basis and raises important issues with the Stockholm Convention ban of POPs. In addition, the identified temporal trends as well as the discussion of the influence of climate change are very valuable.

However, there are some aspects that need to be clarified and improved before publishing the paper. Some aspects concern the communication of results and the respective references to Figures and Tables, I have given some examples of that. Other aspects concern the logical reasoning. Please check this thoroughly.

 I therefore propose this manuscript for acceptance with minor revisions.

Specific comments/ Technical corrections:
Manuscript

Page 2 line 43/P5 l116 – please give a consistent time frame of the investigated studies and check the whole manuscript for it. Here you use 1988-2021 whereas 1980-2021 is given in the title, 1980-2018 in conclusion etc. This should also be consistent with the *years* column given in table 1.

P 3 l 62– Number of substance/substance groups of POPs under SC should be checked, according to my information it should read ´35´.

P5 l18 – please include some more information/criteria on how data quality check of the literature studies was performed.  E.g. did you use a specific data quality check system?

P6 l162 – please correct typing error in trende and table 4)), in addition please check reference to table 4, because it does not exist

P7 l 175 and 180 – There is a confusion about Figures 1 B-D. Fig 1C might be missing, and text in brackets should read 1 B, C and D? If the text is okay, the headline of Fig 1 A-D should be corrected accordingly. In consequence it is difficult to follow this section on HCHs. Please check.

P7 l 186 – A DDTs decreasing trend is not shown in Table 2, Table 2 is on PCBs, please check.

P7 l185-189 – Information on potential spatial differences is missing for DDTs. Is there a reason? Since you discuss spatial differences for some substances it should be included for all substance groups. If not possible, please indicate this in text.

P7 l201 – Please check wording of Sentence ´The estimated half lives …´ because it is not easy to understand. Please also use a consistent wording of *half-lives* in the whole manuscript

P8 l203 – please check reference to Table 4 because it does not exist.

P11 l 315 – delete blank in ( Azcune)

P12 l 332 – Please check meaning of sentence *In response …* it seems that there is something missing.

References:

Please check whole reference section for typing errors, links etc.

Tables:

Why is a table for DDT levels missing such as Tables 1 and 2 for HCBs, HCHs and PCBs? On p5 l121 you refer to the results of the overall literature review but DDT results are not listed. If there is a reason, please indicate.

Table 3 – Please change wording of the heading into ´Estimated atmospheric decreasing times of examined compounds POPs´ because this is the consistent use within the manuscript.

Table 4 - There are several links to Table 4 in the text (see comments above, P9 l246), but Table 4 is missing/does not exist. Please check the whole manuscript.

Figures:

Figure 1 A-D – According to headline 4 figures but only 3 are shown, in addition results might be mixed up in 1A-C, Fig 1C might read 1D and Fig C for b-HCH needs to be included ?? please also check corresponding texts in the whole manuscript

Fig 4 A+B - Please check nomenclature in heading and x-axis legend such as meaning of A-HCH, the use of colours and symbols in figure and x-axis labeling should also be checked, at current stage it is difficult to follow content.

Supplementary Material:

Please check headings in presented tables etc. for typing errors and self-explanatory wording. Please include information on abbreviations e.g. BDL, NR, ND, pp, op ..

---

## Author Comment (AC1)

**Rev 1.**

**Atmospheric Chemistry and Physics, ms -acp-2023-25 General Comments:**

**The manuscript "Levels of persistent organic pollutants (POPs) in the Antarctic atmosphere over time (1980) and estimation of their atmospheric half-lives" by Luarte et al. is well written and of high quality. It reviews scientific literature from 1988 to 2021 that published atmospheric levels of different POPs families. POP families that were studied are polychlorinated biphenyls, hexachlorobenzene, hexachlorocyclohexanes and DDT. With concentration levels published, the respective half-lives were estimated by using characteristic decreasing times. Concentrations levels and half-lives are than compared between the POP families and discussed with their dates of ban under the Stockholm Convention.**

**The detection of POPs in the remote and vulnerable polar environments has been increasingly discussed during the last years. While a lot of studies on the Arctic environment is available, data on POPs occurrence in the Antarctic environment are rare and highly needed. Therefore, this systematic review of POPs in the Antarctic atmosphere provides a good data basis and raises important issues with the Stockholm Convention ban of POPs.**

**In addition, the identified temporal trends as well as the discussion of the influence of climate change are very valuable. However, there are some aspects that need to be clarified and improved before publishing the paper. Some aspects concern the communication of results and the respective references to Figures and Tables, I have given some examples of that.**

**Other aspects concern the logical reasoning.**

**Please check this thoroughly. I therefore propose this manuscript for acceptance with minor revisions.**

We agree with the reviewer about the importance and validity of the results presented in this manuscript, especially due to the lack of studies that integrate the information generated and draw conclusions about the results obtained. The present manuscript is an attempt to systematize this information and analyze what is known about the atmospheric concentrations of these compounds. We have followed the reviewer's recommendations in all his comments and consider that we have answered point by point all the pertinent comments that have definitely improved the text.

Specific comments/ Technical corrections: Manuscript Page 2 line 43/P5 l116 – please give a consistent time frame of the investigated studies and check the whole manuscript for it. Here you use 1988-2021 whereas 1980-2021 is given in the title, 1980-2018 in conclusion etc. This should also be consistent with the years column given in table 1.

We apologize for this mistake now the text in line 43 and 116 says clearly 1980-2021. This was checked also in the whole reviewed version to be consistent.

**P 3 l 62– Number of substance/substance groups of POPs under SC should be checked, according to my information it should read ´35´.**

We checked the number as suggested and we apologize for this mistake thus we changed and followed the recommendation given by the reviewer.

Now the text says:

"Given their detrimental effects, 35 substances/substance groups of POPs are currently regulated internationally by the Stockholm Convention (SC), which seeks to reduce and eliminate POPs production and use (UNECE, 1998; UNEP, 2006)."

**P5 l18 – please include some more information/criteria on how data quality check of the literature studies was performed. E.g. did you use a specific data quality check system?**

We did not use an specific check system but we checked about the blank collected in the field, the reports of limits of detection and quantification, or instrumental detection limits to ensure a correct samples handling and analytical procedures. We expanded this in the corrected version of the manuscript.

Now the text says.

"An exhaustive search was performed in the Web of Science and Scopus databases using the words "Persistent Organic Pollutants", "atmospheric" and "Antarctica", including only articles written in English; excluding from the analysis references that do not refer to a good quality assurance and quality control measures. Thus, studies not reporting information about blank samples, limits of detection, limits of quantification, and/or instrumental detection limits, and referring to previous works reporting the quality criteria used were not included in the present study. This is important because the reported concentrations are at very low levels and to avoid bias it is necessary to be sure about the quality measures of sample collection and analysis."

**P6 l162 – please correct typing error in trend and table 4)), in addition please check reference to table 4, because it does not exist**

We checked this error in table 4 and we apologize for this mistake, now it was corrected, and we checked the references to table 4 as commented by the reviewer. We included the table 4 that was missing in the first version of the manuscript.

| Compounds | $T_{1/2}$ (Years) | 95% Confident Interval | $R^2$ | *p-value* | Equation |
|---|---|---|---|---|---|
| HCB | 14.0 | 10.6-20.7 | 12.03 | <0,0001 | LnCg = -0.04931*year + 100.2 |
| a-HCH | 14.3 | 12.4-17.0 | 32.55 | <0,0001 | LnCg = -0.04817*year + 96.17 |
| y-HCH | 10.1 | 8.6-12.3 | 44.63 | <0,0001 | LnCg = -0.06837*year + 136.9 |
| 4,4´ DDT | 17.2 | 11.8-31.7 | 23.54 | <0,0001 | LnCg = -0.04015*year + 79.50 |
| 2,4 DDT | 14.4 | 9.8-27.3 | 37.55 | <0.001 | LnCg = -0.04794*year + 94.99 |
| 2,4 DDE | 17.6 | 9.2-232 | 15.44 | <0.05 | LnCg = -0.03916*year + 77.93 |
| PCB 28 | 3.9 | 3.2-5.2 | 43.08 | <0,0001 | LnCg = -0.1748*year + 351.2 |
| PCB 52 | 3.7 | 3.2-4.3 | 63.53 | <0,0001 | LnCg = -0.1887*year + 378.7 |
| PCB 101 | 4.7 | 4.0-5.6 | 67.42 | <0,0001 | LnCg = -0.1480*year + 295.8 |
| PCB 118 | 3.6 | 3.0-4.3 | 55.91 | <0,0001 | LnCg = -0.1930*year + 385.8 |
| PCB 138 | 6.5 | 5.3-8.3 | 40.7 | <0,0001 | LnCg = -0.1066*year + 212.7 |
| PCB 153 | 7.6 | 6.0-10.4 | 31.59 | <0,0001 | LnCg = -0.09071*year + 181.2 |
| PCB 180 | 4.6 | 3.3-8.0 | 24.64 | <0,0001 | LnCg = -0.1486*year + 296.2 |

**P7 l 175 and 180 – There is a confusion about Figures 1 B-D. Fig 1C might be missing, and text in brackets should read 1 B, C and D? If the text is okay, the headline of Fig 1 A-D should be corrected accordingly. In consequence it is difficult to follow this section on HCHs. Please check.**

We corrected this aspect in the present version of the manuscript, we corrected the text and the figure.

We corrected the figure heading as follows:

"Figure 1. Atmospheric levels (pg/m$^3$) of HCB (A), -HCH (B), and -HCH (C), over time."

And the corrected figure:

[Figure]

**P7 I 186 – A DDTs decreasing trend is not shown in Table 2, Table 2 is on PCBs, please check.**

We checked this and we included the missing table 2 and we reported the trends in Figure 2. Now

Now the text was corrected as follows:

"Published studies reporting gaseous levels for DDT and their isomers from 1988-2018 were lower than the rest of the target OCPs, and like HCHs, the DDTs showed a decreasing trend over the years (Table 2, Fig. 2), with significant inter-annual differences (p<0.05) for compounds 4,4'-DDT, 4,4'-DDE, 2,4'-DDT and 2,2'-DDE, and non-significant annual differences (p>0.05) for compounds 4,4'-DDD and 2,4'-DDD."

And here we reproduce the Table 2and figure 2

| Sampling area | Type of sampling | Year | 2,4 DDE | 4,4-DDE | 2,4DDD | 4,4 DDD | 2,4 DDT | 4,4 DDT | Reference |
|---|---|---|---|---|---|---|---|---|---|
| Ross Island | Passive | 1988 - 1999 | | 1 | | | | 2 | Larson et al., 1992 |
| East Antarctica | Passive | 1990 | | | | | | 0.53 | Bidleman et al., 1993 |
| Signy Island | Active | 1994 - 1995 | 0.07 | 0.4 | 0.068 | 0.098 | 0.195 | 0.2 | Kallenborn et al., 1998 |
| Ross Island | Active | 1995 | | 9.2 | | 11.7 | | 8.1 | Montone et al., 2005 |
| Antarctic marginal seas | Active | 2013-2014 | 0.097 | 0.35 | 0.043 | 0.034 | 0.17 | 0.12 | Wu et al., 2020 |
| Southern Ocean between Australia and Antarctica | Active | 2014 | <0.51 | <0.15-0.44 | <1,6 | <1,8 | <2,7 | <7,8 | Bigot et al., 2016 |
| King George Island | Paasive | 2012-2018 | 0.2 | 0.6 | 0.1 | 0.2 | 0.1 | 0.24 | Hao et al., 2019 |

And the figure:

[Figure]

[Figure]

[Figure]

**P7 l185-189 – Information on potential spatial differences is missing for DDTs. Is there a reason? Since you discuss spatial differences for some substances it should be included for all substance groups. If not possible, please indicate this in text.**

We expanded this in the text as follows to clarify the text.

"The detected concentrations of HCB, -HCHs, and 4,4'-DDT indicate significant spatial differences (P<0.05), with higher atmospheric concentrations in West Antarctica than in East Antarctica (Table S.4). The -HCH, and 2,4´-DDT isomers did not show spatial differences between the two zones (P>0.05) (Table S.4). The U-Mann Whitney variance analysis was not performed for the -HCH and 2,4'-DDD isomers, because all levels reported in East Antarctica were below the detection limit. In addition, the results of the Generalized Linear Model indicate that the variability of atmospheric OCPs is mainly due to the year variable, with no significant differences (p>0.05) between the atmospheric levels obtained from active and passive sampling (see Table S.5)."

**P7 l201 – Please check wording of Sentence ´The estimated half lives …´ because it is not easy to understand. Please also use a consistent wording of half-lives in the whole manuscript P8 l203 – please check reference to Table 4 because it does not exist.**

We corrected this in the whole manuscript as suggested both the use of atmospheric half live nomenclature and the table 4.

**P11 l 315 – delete blank in ( Azcune)**

We corrected this typo.

**P12 l 332 – Please check meaning of sentence In response … it seems that there is something missing. References: Please check whole reference section for typing errors, links etc.**

We checked the meaning of the sentence as suggested by the reviewer and we corrected the whole sentence. We reviewed the section and now the text says.

Now the text says

"As presented here, several factors can be considered as sources of bias from historical data analysis. First, in the time frame of this study (1980-2021), analytical instrumentation and laboratory techniques exhibited dramatic change, particularly with the advent of advanced mass spectrometry (MS) over electron capture detection (ECD) or novel calibration techniques based on isotopically labeled standards (Azcune et al., 2022). Therefore, recent data are generated by more sophisticated. On the other hand, we also included studies using active and passive sampling, but no major differences in the values obtained were observed (See Tables 1, 2, and 3) for the whole compounds which agrees with intercalibration experiments conducted in other areas comparing passive and active sampling together (Prats et al., 2022).
The published information from Antarctica is reduced to a group of individual experiences in different geographical locations of international teams working in the field under different conditions and levels of competence that are difficult to obtain and analyze. One might expect that studies that show a strong track record in Antarctic research, reporting POP levels over a time series, might have greater validity due to constancy and consistency in both sampling and the types of analyses used (e.g., Larson et al., 1992; Baek et al., 2011; Hao et al., 2019).
In summary, the source of bias, related to the technological advancement of the analyses of the collected samples, could have relevance in the observed variability of the historical trends of HCB and HCH (see Fig. 1 A to C). In these cases, it is suggested to continue with dedicated monitoring of these POPs in the coming years to obtain robust observations and conclusions on the degradation of POPs in the Antarctic atmosphere.

**Tables: Why is a table for DDT levels missing such as Tables 1 and 2 for HCBs, HCHs and PCBs? On p5 l121 you refer to the results of the overall literature review but DDT results are not listed. If there is a reason, please indicate.**

The table of atmospheric levels of DDT was added to the manuscript and was missing there is no reason to not show this table. Now we included this table in the revised version of the manuscript. We apologize for this mistake, and we are grateful for the reviewer´s comment.

Here we reproduce the table

Table 2. DDTs levels (pg/m$^{-3}$) in Antarctic atmosphere since 1988 to present.

| Sampling area | Type of sampling | Year | 2,4 DDE | 4,4-DDE | 2,4DDD | 4,4 DDD | 2,4 DDT | 4,4 DDT | Reference |
|---|---|---|---|---|---|---|---|---|---|
| Ross Island | Passive | 1988 - 1999 | | 1 | | | | 2 | Larson et al., 1992 |
| East Antarctica | Passive | 1990 | | | | | | 0.53 | Bidleman et al., 1993 |
| Signy Island | Active | 1994 - 1995 | 0.07 | 0.4 | 0.068 | 0.098 | 0.195 | 0.2 | Kallenborn et al., 1998 |
| Ross Island | Active | 1995 | | 9.2 | | 11.7 | | 8.1 | Montone et al., 2005 |
| Antarctic marginal seas | Active | 2013-2014 | 0.097 | 0.35 | 0.043 | 0.034 | 0.17 | 0.12 | Wu et al., 2020 |
| Southern Ocean between Australia and Antarctica | Active | 2014 | <0.51 | <0.15-0.44 | <1,6 | <1,8 | <2,7 | <7,8 | Bigot et al., 2016 |
| King George Island | Paasive | 2012-2018 | 0.2 | 0.6 | 0.1 | 0.2 | 0.1 | 0.24 | Hao et al., 2019 |

**Table 3 – Please change wording of the heading into ´Estimated atmospheric decreasing times of examined compounds POPs´ because this is the consistent use within the manuscript.**

We reviewed the whole manuscript to be consistent and coherent in the nomenclature used and the changed the term atmospheric decreasing times to estimated atmospheric half-life, which is more consistent with our estimation methodology.

**Table 4 - There are several links to Table 4 in the text (see comments above, P9 l246), but Table 4 is missing/does not exist. Please check the whole manuscript.**

We added the table of atmospheric levels of DDT (Table 2). We were leaving a total of 4 tables. It was verified that the citations corresponded to the tables. See previous comment regarding this.

**Figures: Figure 1 A-D – According to headline 4 figures but only 3 are shown, in addition results might be mixed up in 1A-C, Fig 1C might read 1D and Fig C for b-HCH needs to be included ?? please also check corresponding texts in the whole manuscript**

We agree with reviewer´s comments, and we decided to eliminate b-HCH following the recommendation. In fact, data of b-HCH were scarce and the figure did not give to much information and is impossible to estimate a trend based on the small amount of published data.

We corrected this with same action done for a previous comment.

We corrected the figure heading as follows:

"Figure 1. Atmospheric levels (pg/m$^3$) of HCB (A),  -HCH (B), and  -HCH (C), over time."

And the corrected figure:

[Figure]

The figure heading was corrected by eliminating figure D. Together with the co-authors, it was decided to eliminate the figure for b-HCH due to the limited published data for this compound.

**Fig 4 A+B - Please check nomenclature in heading and x-axis legend such as meaning of A-HCH, the use of colours and symbols in figure and x-axis labeling should also be checked, at current stage it is difficult to follow content.**

We agree with the reviewer's suggestion and have improved both the figure and the legend. We believe that now the legend of the figure is clearer and the figure can be better understood.

Now the legend says:

"Figure 4. Comparison among the estimated atmospheric half-lives obtained in the present work (green) compared with similar estimations from the Great Lakes (red; Vernier and Hites, 2010) and the Arctic (blue; Wong et al., 2021) for A) Organochlorine pesticides (HCB, α-HCH, γ-HCH, 2,4'-DDT and 4,4'-DDT) and B) Polychlorinated biphenyls (28, 52, 101, 118, 138, 153 and 180), A-HCH, DDX, and B) PCBs"

And the new arranged figure seems easier to follow:

[Figure]

**Supplementary Material: Please check headings in presented tables etc. for typing errors and self-explanatory wording. Please include information on abbreviations e.g. BDL, NR, ND, pp, op** ..

Below tables S.1. and S.2. of the supplementary material the meaning of the acronyms present in each table is specified.

Now the headings of tha tables say:

"Table S.1. Reported atmospheric levels for the OCPs isomers reviewed. HCB (Hexachlorobenzene), α-HCH ( a isomer of hexachlorocyclohexane, b-hch (b isomer of hexachlorocyclohexane), y-HCH (y isomer of hexachlorocyclohexane), 4,4'-DDT (4 4' diclorodyphenyl trichloroethane), 2,4'-DDT ( 2 4' diclorodyphenyl trichloroethane), 4,4' DDE (4, 4' Dichlorodyphenyl dichloroethylene), 2,4 (2,4 Dichlorodyphenyl dichloroethylene), 4,4' DDD (4,4' diclorodyphenyl dichloroethane), 2,4' (2,4'- diclorodyphenyl dichloroethane). ND: Not detected, NR: Not reported, BDL: Below detection limite, LOQ: Limit of quantification"

"Table S.2. Reported atmospheric levels for the 7 polychlorinated byphenyls (PCBs) congeners reviewed. ND: Not detected, NR: Not reported, BDL: Below detection limite, LOQ: Limit of quantification"

Rev 2

**General Comments**

**This manuscript reviewed literatures for the atmospheric concentrations of several classes POPs in the Antarctica from 1988 to 2021. Temporal trends were evaluated for DDT, DDD PCBs, HCHs and HCB according to the effective ban of SC. Atmospheric half-life times of these POPs were estimated using characteristic decreasing times (TD).**

**The results showed that the ban of SC significantly influenced the levels of HCHs and PCBs, while HCB showed increasing concentrations in some publications, and longer half-life time than other POPs.**

**The impact of climate change on the POPs levels was discussed. Increasing temperature can cause remission of POPs from the surface, and other biogeochemical processes. Overall, the manuscript have been well documented, and addressed to the emerging concern for POPs in the Antarctic. I would suggest it can be accepted with some revision.**

We agree with the reviewer in their comments provided below and we corrected the manuscript following their valuable comments that we think improved the text in a significant way.

**Specific comments**

**L116-118, including only articles written in English; excluding from the analysis references that do not refer to a good quality assurance and quality control during the chemical analysis, or if the levels of field blanks were not reported.**
**Please give more detail description for "good quality assurance and quality control during the chemical analysis" applied for literature selection**

We agree with this comment, English language is not a decisive issue to check the quality criteria for selection of the studies thus we expanded our criteria in the text giving more details.

Now the text says:

"An exhaustive search was performed in the Web of Science and Scopus databases using the words "Persistent Organic Pollutants", "atmospheric" and "Antarctica", including only articles written in English; excluding from the analysis references that do not refer to a good quality assurance and quality control measures. Thus, studies not reporting information about blank samples, limits of detection, limits of quantification, and/or instrumental detection limits, and referring to previous works reporting the quality criteria used were not included in the present study. This is important because the reported concentrations are at very low levels and to avoid bias it is necessary to be sure about the quality measures of sample collection and analysis"

**L121-122, data obtained from active and passive sampling**

**As data from both active and passive sampling were collected in this work, although the authors stated no clear variation between these two data sets, I guess it is worth to compare the data between active sampling and passive sampling in this review, and give a suggestion for future monitoring program.**

We agree with this comment, so we decided to include a GLM in the analysis considering the terms time (year) and method (Active vs passive). The results of the analysis indicated that the atmospheric POPs concentration obtained, and the variation was only related to time and no influence of the sampling methodology was identified. indicate that the variability of atmospheric POP concentrations is mainly due to the years and not to the type of sampling used (Table S.5.). On the other hand we could suggest that the sampling strategy should be selected according to the objectives of the study design. For example, if our focus in a study is a high-resolution monitoring and no logistic restriction is found (i.e electricity) them the sampling strategy for

monitoring should be active but this will limit the spatial cover of the study. On the other hand, if a highly geographical coverage is needed, less time resolution and logistics are complicated (no electricity available) then passive sampling will be the correct option. However, we think that this is far from the objectives of the present study. But we included a comment on this aspect in the source of bias section.

New text could be found in

Methodology section, page 5, line 130-133.

"Finally, a Generalized Linear Model (GLM) was performed to elucidate whether the variability in the atmospheric POP levels reviewed is due to the different types of sampling used by the different studies (active or passive sampling) or the time variable."

OCP results, page 7, lines 200-202.

"2,4'-DDD isomers, because all levels reported in East Antarctica were below the detection limit. In addition, the results of the Generalized Linear Model indicate that the variability of atmospheric OCPs is mainly due to the year variable, with no significant differences (p>0.05) between the atmospheric levels obtained from active and passive sampling (see Table S.5)."

PCB results, pages 9 and 10, lines 263-266.

". On the other hand, it is essential to highlight that the variability of PCBs reported in this study is substantially due to the time variable (p<0.05), with no significant differences (p>0.05) between the atmospheric levels of PCBs obtained from active and passive sampling (see Table S.5)."

And Potential source of bias section, page 10-11, lines 322-326

"Therefore, recent data are generated by more sophisticated techniques and modern laboratory QA/QC criteria. On the other hand, we also included studies using active and passive sampling, but no major differences in the values obtained were observed (See Tables 1, 2, and 3) for the whole compounds which agrees with intercalibration experiments conducted in other areas comparing passive and active sampling together (Prats et al., 2022)."

Also in the section conclusions we commented out thoughts regarding the use of active and passive sampling for future monitoring networks.

Now the text says:

"There is increasing evidence of the presence of emerging compounds in different environmental matrices in Antarctica, however, the current surveillance of atmospheric pollutants is related to specific research groups, instead of coordinated efforts between countries with Antarctic presence, where continuous monitoring networks could be generated with the inclusion of various persistent toxic chemicals, as analogous to the efforts done by the Arctic Monitoring and Assessment Program (AMAP), or the Integrated Atmospheric Deposition Network (IADN) in the Great Lakes. In this sense to stablish a monitoring program for assessment of POPs levels in Antarctic Atmosphere will depend on the capabilities and the facilities, since active sampling strategy will benefit a higher resolution in the assessment of POPs trends in the monitoring points but on the other hand the use of passive sampling strategy could represent a high spatial coverage to monitor trends but lower time resolution. But there is need to stablish in the future a bigger monitoring network coordinated following the previous experience gained from AMAP."

**L189-197, spatial distribution of HCB, a-HCH, b-HCH, and g-HCH isomers was discussed in this section. Please give more discussion for the significant spatial differences of HCB and a-HCHs between East and West Antarctica.**

We discussed this in the text however is difficult to discuss thus topic due to the lack if information we included a paragraph trying to summarize the information. However we only could argue that the proximity of South America to Western Antarctica and the data on historical usage and emissions from continents in the bibliography could suggest and explanation to this differences:

Now the text says:

"The detected concentrations of HCB, -HCH, and 4,4'-DDT indicate significant spatial differences (P<0.05), with higher atmospheric concentrations in West Antarctica than in East Antarctica (Table S.4). The -HCH, and 2,4´-DDT isomers did not show spatial differences between the two zones (P>0.05) (Table S.4), but the usage of this compounds decreased in a similar way from 1990 to 2000 (Vijgen, 2006). This can be explained by two causes together, the first is the greater proximity of South America to the Antarctic Peninsula. The proximity itself has to do with the possibility of transport of these compounds from southern South America where it is suggested that air samples influenced by the continent are capable of transporting pollutants from South America to Antarctica (Dickhutt et al., 2005) such as Heptachlor epoxide. However, when looking to usage reported in South America compared to Africa (Li, 1999). Thus, this could influence the abundance of -HCH in the western Antarctic area. Examining previous information for both HCB and 4,4'-DDT there is not a great deal of information about the use of these compounds in areas near Antarctica but the proximity to South America could explain these variations in conjunction with the paucity of data in Eastern Antarctica. On the other hand, the U-Mann Whitney variance analysis was not performed for the -HCH and 2,4'-DDD isomers, because all levels reported in East Antarctica were below the detection limit. In addition, the results of the Generalized Linear Model indicate that the variability of atmospheric OCPs is mainly due to the year variable, with no significant differences (p>0.05) between the atmospheric levels obtained from active and passive sampling (see Table S.5)."